# Peptide-Based Vaccine against SARS-CoV-2: Peptide Antigen Discovery and Screening of Adjuvant Systems

**DOI:** 10.3390/pharmaceutics14040856

**Published:** 2022-04-13

**Authors:** Ahmed O. Shalash, Armira Azuar, Harrison Y. R. Madge, Naphak Modhiran, Alberto A. Amarilla, Benjamin Liang, Alexander A. Khromykh, Waleed M. Hussein, Keith J. Chappell, Daniel Watterson, Paul R. Young, Mariusz Skwarczynski, Istvan Toth

**Affiliations:** 1School of Chemistry and Molecular Biosciences, The University of Queensland, St Lucia, QLD 4072, Australia; a.shalash@uqconnect.edu.au (A.O.S.); armira.azuar@uq.net.au (A.A.); harrison.madge@uq.net.au (H.Y.R.M.); n.modhiran@uq.edu.au (N.M.); a.amarillaortiz@uq.edu.au (A.A.A.); benjaminjianwen.liang@uq.net.au (B.L.); alexander.khromykh@uq.edu.au (A.A.K.); w.hussein@uq.edu.au (W.M.H.); k.chappell@uq.edu.au (K.J.C.); d.watterson@uq.edu.au (D.W.); p.young@uq.edu.au (P.R.Y.); 2Australian Institute for Bioengineering and Nanotechnology, The University of Queensland, St Lucia, QLD 4072, Australia; 3School of Pharmacy, The University of Queensland, Woolloongabba, QLD 4102, Australia

**Keywords:** SARS-CoV-2, nanovaccine, live virus neutralization, pseudovirion neutralization, ACE2 binding inhibition, subcutaneous immunization, peptide vaccine, receptor binding domain

## Abstract

The SARS-CoV-2 virus has caused a global crisis, resulting in 0.5 billion infections and over 6 million deaths as of March 2022. Fortunately, infection and hospitalization rates were curbed due to the rollout of DNA and mRNA vaccines. However, the efficacy of these vaccines significantly drops a few months post immunization, from 88% down to 47% in the case of the Pfizer BNT162 vaccine. The emergence of variant strains, especially delta and omicron, have also significantly reduced vaccine efficacy. We propose peptide vaccines as a potential solution to address the inadequacies of the current vaccines. Peptide vaccines can be easily modified to target emerging strains, have greater stability, and do not require cold-chain storage. We screened five peptide fragments (B1–B5) derived from the SARS-CoV-2 spike protein to identify neutralizing B-cell peptide antigens. We then investigated adjuvant systems for efficient stimulation of immune responses against the most promising peptide antigens, including liposomal formulations of polyleucine (L10) and polymethylacrylate (PMA), as well as classical adjuvants (CFA and MF59). Immune efficacy of formulations was evaluated using competitive ELISA, pseudovirion neutralization, and live virus neutralization assays. Unfortunately, peptide conjugation to L10 and PMA dramatically altered the secondary structure, resulting in low antibody neutralization efficacy. Of the peptides tested, only B3 administered with CFA or MF59 was highly immunogenic. Thus, a peptide vaccine relying on B3 may provide an attractive alternative to currently marketed vaccines.

## 1. Introduction

The COVID-19 pandemic pandemic has been a major healthcare disaster. Since the first case was documented in December 2019 until now (February 2022), the virus has caused 415 million infections and 5.8 million global deaths. COVID-19 infection occurs through the inhalation of contaminated aerosol from an infected patient. SARS-CoV-2 (SARS-2) binds via its spike protein (SARS-2-S) to the angiotensin-converting enzyme-2 (ACE2) [1]. Binding changes the conformation of SARS-2-S and allows priming by human host proteases to reveal the fusion peptide within its sequence. SARS-2 fuses with the host cell and viral (+) ssRNA is translated into a polyprotein precursor that is further cleaved to yield functional structural and nonstructural proteins to replicate the virus [2,3,4,5].

A vaccine’s efficacy is defined by its ability to induce neutralizing and opsonizing humoral responses to protect a host from infection, and its ability to trigger cytotoxic T-cell (CTL) responses against infected cells to clear existing infection [6]. Analysis of SARS-2 convalescent patient serum has shown that most antibodies are directed against SARS-2-S and 90% of the “neutralizing” antibodies are directed against the receptor binding domain (RBD) in SARS-2-S [7,8,9,10]. Neutralizing antibodies were found to be protective at very low concentrations (7 ng/mL) in the serum [11]. Recently, a correlation was reported between anti-RBD IgG titers and viral titer reductions in rhesus monkeys infected with SARS-2 [12,13]. Thus, identifying the epitope footprints of such potent neutralizing antibodies could underpin the development of a highly effective peptide vaccine antigen [14]. Most relevant epitopes are located in the receptor binding motif (RBM) domain within the RBD; however, some are highly discontinuous and their epitope footprints extend into *C*-terminal residues, *N*-terminal residues of the RBM, or even outside of the RBM sequence [15,16,17,18]. Still, the RBM sequence is a logical target for deriving peptide vaccine antigens.

Current vaccine antigens are predominantly genetic: RNA-encoded SARS-2-S (mRNA-1273, Moderna), DNA-encoded SARS-2-S (ChAdOx-1 nCoV-19, Astra Zeneca), or RNA-encoded RBD (BNT162b1, Pfizer), encapsulated in lipid transfection delivery systems (mRNA-1273 and BNT162b1) [19,20] or live, nonreplicating liver adenovirus vector (ChAdOx-1 nCoV-19) [21]. Live viruses have high transfection efficacy; however, they may cause considerable adverse systemic events. Although nonlive lipoplexes are safer, they may also cause adverse effects [19,20,21]. Indeed, the genetic vaccines have triggered a variety of side effects across the vaccinated population. The risk of side effects can also cause a fear of vaccination and, consequently, may have contributed to suboptimal vaccination rates in some countries [22,23]. In addition, not all epitopes within protein sequences are safe; some can cause immunopathological symptoms [24,25,26]. Importantly, the currently approved COVID vaccines (a) tend to rapidly lose their protective efficacy with time; e.g., the protective efficacy of BNT-162 against infection dropped from 88% to 47% over 5 months [27]; (b) are less effective against emerging variants, as highlighted particularly by emergence of the omicron and delta strains [28,29,30]; and (c) face several difficulties in relation to scaling up, storage and transport under strict cryotemperatures. Further limitations arise due to a lack of manufacturing facilities equipped to prepare genetic vaccines. These issues restrict pandemic-scale rollout of genetic vaccines and create the need for alternative approaches; vaccines based on minimal neutralizing epitopes present an attractive alternative. Consequently, the RBD, especially the RBM sequence, provides a promising target with several potential neutralizing epitopes. The generation of immune responses against these should block binding between ACE2, the main host cell target receptor, and the virus [10,31,32].

Peptide vaccines have been proven to be highly immunogenic when properly adjuvanted [33,34]. They can be designed to omit immunopathological pro-inflammatory sequences, focus immune response on neutralizing epitopes, and combine antigens from different viral proteins [35]. Peptide vaccines are also easy to chemically synthesize, process, and purify, are scalable for mass production to meet current global demand, and are devoid of biological contaminants [36]. Peptide vaccines that contained RBD-derived antigens against SARS-CoV-2 have recently been shown to trigger humoral neutralizing responses [14,35,37,38]. Moreover, peptide antigens can be modified to eliminate the need for an adjuvant. For example, polyleucine- and polymethyl acrylate (PMA)-conjugated peptide antigens demonstrated excellent immunogenicity alone and in combination with liposomal delivery systems [34,39,40]. Skwarczynski et al. demonstrated that polyleucine conjugated to a peptide antigen derived from Group A Streptococcus (GAS) M protein self-assembled into small nanoparticles 10–30 nm in diameter that were able to trigger production of high levels of antigen-specific antibodies and clear GAS infection following subcutaneous immunizations in mice [34]. Similarly, PMA conjugated to the same antigen or gonadotropin hormone-releasing hormone generated high levels of antibody production even after a single oral immunization [41,42]. Moreover, liposomes loaded with PMA conjugated to a human papilloma virus-derived peptide antigen triggered strong immune responses, which were able to eradicate a model cervical cancer in mice, while classical adjuvant (Montanide ISA 51) failed to induce adequate immunity [39,40]. Thus, peptide-based vaccines could potentially overcome the hurdles in rollout that the current genetic vaccines have experienced, while remaining safe, tolerable, and effective. In addition, peptide vaccines may provide high feasibility, avoidance of cold-chain complexity, easier modification for emerging strains, greatly improved stability, and scalable synthesis and production processes that are free from biocontaminants.

We aimed to initially screen the RBD for immunogenic and neutralizing epitopes to employ as minimal antigenic components in peptide-based vaccines to generate an effective neutralizing immune response. We then synthesized the promising peptide antigens with self-adjuvating moieties, such as polyleucine or poly(methyl acrylate), as potential self-adjuvanted peptide-based vaccines [34,43]. The developed self-adjuvanted peptide vaccines were loaded into delivery systems comprising liposomes. Human use-approved commercial adjuvant, MF59-, and gold-standard complete Freund’s adjuvant (CFA)-adjuvanted antigens were used as a positive control. The antibodies generated were screened for their neutralization potential using several sensitive in vitro tests, including competitive ELISA, pseudovirion neutralization assay, and live virus neutralization assay.

## 2. Materials and Methods

### Materials

Fluorenylmethyloxycarbonyl (Fmoc)-protected l-amino acids, 4-methylbenzhydrylamine (MBHA), Rink amide, polymer-bound, was purchased from Novabiochem (Läufelfingen, Switzerland) and from Mimotopes (Melbourne, Australia). The reagent 2-(1*H*-7-azabenzotriazol-1-yl)-1,1,3,3-tetramethyluronium hexafluorophosphate (HATU) was purchased from Mimotopes, while *N*,*N*-diisopropylethylamine (DIEA), *N*,*N*-dimethyl formamide (DMF), trifluoroacetic acid (TFA), diethyl ether, dichloromethane (DCM), HPLC gradient grade acetonitrile (ACN) LiChrosolv^®^, methanol, and piperidine were purchased from Merck (Darmstadt, Germany). 4-pentynoic acid was obtained from Novachem Pty Ltd. (Victoria, Australia). Triisopropylsilane (TIS), bovine serum albumin (BSA) electrophoresis grade 98% pure, goat antihuman IgG (Fc specific)-peroxidase, and all other materials as analytical purity grade, were obtained from Sigma-Aldrich (Castle Hill, Australia). Fc-ACE2 protein (human, recombinant ACE2 with human IgG-Fc tag) and Wuhan-1 original strain SARS-2-RBD (position: S^319^–S^541^, 2019-nCoV spike RBD-His recombinant protein) were obtained from Sino Biologicals (Beijing, China, Cat. No. 10108-H02H).


**Synthesis and Purification of Peptide Antigens**


Antigen and conjugate synthesis were conducted using Fmoc-based solid-phase peptide synthesis (Figure 1). Resin was swelled in DMF, followed by cycles of double deprotection using piperidine 20% in DMF, and double couplings using a mixture of activated Fmoc-protected amino acids (4.2 equivalents), HATU (4.0 equivalents), and DIEA (6.2 equivalents) in DMF solution. This was done to couple each amino acid to the sequence until the sequence was complete. The *N*-terminus was then acetylated, using acetic anhydride and DIEA in DMF (0.5:0.5:9.0), and the resin was dried overnight in a desiccator. Thereafter, the peptide was cleaved from the resin using a 20 mL/g resin cleavage cocktail solution of TIS, MilliQ water, and TFA (0.25:0.25:9.5). Following cleavage, the peptides were precipitated using cold diethyl ether, extracted using a mixture of acetonitrile, MilliQ water and TFA (45:55:0.1), and freeze-dried to yield crude dry peptide. Where relevant, disulfide bonds were formed post-cleavage, as recently reported [44], by dissolving sulfhydryl peptides in 0.1 M ammonium bicarbonate buffer solution at 1 mg/mL concentration. The solution was allowed to oxidize by stirring for 3 days, followed by freeze-drying. Disulfide bond formation was confirmed by mass spectrometry and reverse-phase liquid chromatography (RP-HPLC).

Crude compounds were analyzed using analytical Shimadzu HPLC (Shimadzu Corp., Kyoto, Japan) equipped with a Vydac C4 (214TP54; 5 mm, 4.6 × 250 mm) or C18 column (218TP54; 5 mm, 4.6 × 250 mm) at a 1 mL/min flowrate and 214 nm detection wavelength (Appendix A). Separation was achieved by a gradient program employing an organic modifier mobile phase solvent B (ACN, TFA, and MilliQ water with volume ratio of 90:0.1:10) increased from 0% to 100% over 40 min at the expense of solvent A (TFA in MilliQ water with a volume ratio of 0.1:100). Identification was conducted by injecting 5 μL of 1 mg/mL compound solution into electrospray ionization mass spectrometry (ESI-MS; Sciex API3000, MDS Sciex, Toronto, ON, Canada) and matching the observed/found spectrum to the calculated one using Chemdraw Professional version 16.0 (Perkin Elmer, MA, USA). Purification was conducted using preparative RP-HPLC (Shimadzu Corp., Kyoto, Japan) with a Vydac C4 (214TP1022; 10 mm, 22 × 250 mm) or C18 (218TP1022; 10 mm, 22 × 250 mm) preparative column and a flowrate of 20 mL/min and detection wavelength of 214 nm. Separation was achieved using a partial gradient program, similar to that of analytical RP-HPLC, which scales up the chromatogram region of interest to isolate the pure compound. Pentynoylated-B3 or pentynoylated-PADRE were conjugated to azide-terminated polymethyl acrylate polymer, as described previously [45], and RP-HPLC and MALDI-ToF were used to evaluate conjugate purity and identity, respectively (Appendix A). Peptide antigen sequences and their locations on the original Wuhan-1 strain SARS-CoV-2 spike protein sequence are demonstrated on Figure 1 and Appendix A.

B1 peptide (^623^AIHADQLTPTWRVYSTG^639^) at the S1/S2 cleavage/priming site. Yield: 75%. Molecular weight: 1957.18. ESI-MS: [M + 2H]^2+^ *m*/*z* 979.6 (calc. 979.5), [M + 3H]^3+^ *m*/*z* 653.5 (calc. 653.39). t_R_ = 17.97 min (0 to 100% solvent B; C18 column). Purity = 98%.

B2 peptide (^469^STEIYQAGSTP**C**NGVEGFN**C**YFPLQSYGFQPTNGVGYQPY^508^) at the RBM of RBD. Yield: 45%. Molecular weight: 4425.8. ESI-MS: [M + 3H]^3+^ *m*/*z* 1476.8 (calc. 1476.27), [M + 4H]^4+^ *m*/*z* 1108.0 (calc. 1107.45). t_R_ = 21.15 min (0 to 100% solvent B; C18 column). Purity ≥ 99%.

B3 peptide (^444^GVGGNYNYLYRLFRKSNLKPFERDISTEIYQAGSTPCNGV^483^) at the RBM of RBD. Yield: 50%. Molecular weight: 4593.2. ESI-MS: [M + 3H]^3+^ *m*/*z* 1532.3 (calc. 1532.07), [M + 4H]^4+^ *m*/*z* 1149.10 (calc. 1149.30), [M + 5H]^5+^ *m*/*z* 919.5 (calc. 919.64). t_R_ = 21.49 min (0 to 100% solvent B; C18 column). Purity ≥ 99%.

B4 peptide (^559^FLPFQQFGRDIADT^572^) near the S1/S2 cleavage/priming site. Yield: 80%. Molecular weight: 1675.8. ESI-MS: [M + 1H]^1+^ *m*/*z* 1676.8 (calc. 1676.8), [M + 2H]^2+^ *m*/*z* 839.4 (calc. 839.9), [M + 3H]^3+^ *m*/*z* 559.9 (calc. 559.6). t_R_ = 19.35 min (0 to 100% solvent B; C18 column). Purity ≥ 99%.

B5 peptide (^366^SVLYNSASFSTFKCYGVSPTKLNDLCFTNV^395^) at the *N*-terminus of RBD, but not in direct contact with ACE2. Yield: 55%. Molecular weight: 3348.6. ESI-MS: [M + 2H]^2+^ *m*/*z* 1675.2 (calc. 1675.3), [M + 3H]^3+^ *m*/*z* 1117.1 (calc. 1117.2), [M + 4H]^4+^ *m*/*z* 838.6 (calc. 838.15). t_R_ = 17.2 min (0 to 100% solvent B; C18 column). Purity ≥ 99%.

PADRE peptide (AKFVAAWTLKAAA). Yield: 75%. Molecular weight: 1389.7. ESI-MS: [M + 1H]^1+^ *m*/*z* 1390.7 (calc. 1390.7), [M + 2H]^2+^ *m*/*z* 695.8 (calc. 695.9). t_R_ = 23.2 min (0 to 100% solvent B; C18 column). Purity ≥ 99%.

L_10_-B1 peptide (L_10_-AIHADQLTPTWRVYSTG). Yield: 50%. Molecular weight: 3088.8. ESI-MS: [M + 2H]^2+^ *m*/*z* 1545.4 (calc. 1545.3), [M + 3H]^3+^ *m*/*z* 1030.6 (calc. 1030.5). t_R_ = 17.97 min (0 to 100% solvent B; C4 column). Purity = 96%.

L_10_-B2 (L_10_-STEIYQAGSTP**C**NGVEGFN**C**YFPLQSYGFQPTNGVGYQPY). Yield: 35%. Molecular weight: 5560.4. ESI-MS: [M + 3H]^3+^ *m*/*z* 1854.5 (calc. 1854.4), [M + 4H]^4+^ *m*/*z* 1391.1 (calc. 1391.1), [M + 5H]^5+^ *m*/*z* 1113.1 (calc. 1113.1). t_R_ = 30.2 min (0 to 100% solvent B; C4 column). Purity ≥ 99%.

L_10_-B3 (L_10_-GVGGNYNYLYRLFRKSNLKPFERDISTEIYQAGSTPCNGV). Yield: 40%. Molecular weight: 5747.40. ESI-MS: [M + 4H]^4+^ *m*/*z* 1437.2 (calc. 1437.85), [M + 5H]^5+^ *m*/*z* 1149.8 (calc. 1150.48), [M + 6H]^6+^ *m*/*z* 958.9 (calc. 958.90). t_R_ = 25.64 min (0 to 100% solvent B; C4 column). Purity ≥ 99%.

L_10_-PADRE (L_10_-AKFVAAWTLKAAA). Yield: 45%. Molecular weight: 2521.27. ESI-MS: [M + 2H]^2+^ *m*/*z* 1260.6 (calc. 1261.64), [M + 3H]^3+^ *m*/*z* 841.0 (calc. 841.42). t_R_ = 37.10 min (0 to 100% solvent B; C4 column). Purity ≥ 99%.

PMA-B3 (PMA-GVGGNYNYLYRLFRKSNLKPFERDISTEIYQAGSTPCNGV). Yield: 51%. Molecular weight: 6.9 ± 0.3 kDa. MALDI-ToF: [M + 1H]^1+^ *m*/*z* 6.8 kDa. t_R_ = 33.50 min (25 to 100% solvent B; C4 column). Purity = 97%.

PMA-PADRE (PMA-AKFVAAWTLKAAA). Yield: 43%. Molecular weight: 3.75 ± 0.3 kDa. MALDI-ToF: [M + 1H]^1+^ *m*/*z* 3.77 kDa. t_R_ = 33.2 min (25 to 100% solvent B; C4 column). Purity = 98%.


**Secondary Structure of Peptide Antigens**


Antigen secondary structure was evaluated by preparing a 1 mg/mL solution of each synthesized peptide in phosphate buffered saline (PBS). Aliquots of 0.2 mL were transferred into 1 mm pathlength quartz microcuvettes and measured via circular dichroism (CD) spectra using a Jasco-720 spectropolarimeter (Jasco Corp., Tokyo, Japan). Quantitative secondary structure content was determined by fitting the composite spectra to yield contribution of each pure individual component spectrum, i.e. α-helix, β-sheet, or random coil, obtained from reported polylysine spectra, adapted from the method introduced by Greenfield et al. [46,47], as described previously [48].


**Liposome Preparation and Encapsulation Efficiency**


Liposomes were prepared by dissolving and mixing dipalmitoylphosphatidylcholine (DPPC, 5 mg, 6.8 × 10^−3^ mmole), didodecyldimethylammonium bromide (DDAB, 2 mg, 5.2 × 10^−3^ mmole), and cholesterol (1 mg, 8.3 × 10^−3^ mmole) in 2 mL of chloroform in a 5 mL round-bottom flask. Polyleucine- or PMA-conjugated antigen peptide, single (0.6 mg) or double dose (1.2 mg), along with helper T-cell epitopes L_10_-PADRE or PMA-PADRE (0.6 mg), were dissolved in 2 mL of methanol and chloroform mixture (1:1). This was then added to the chloroform lipid solution in the round-bottom flask. The flask was attached to a rotary evaporator (Büchi, Switzerland) and the mixture was evaporated under reduced pressure (95 mbar) for 2 h at 30 °C. The formed thin layer of film was left to dry overnight in a desiccator under vacuum to remove any residual solvents. The dry lipid film was then allowed to rehydrate and swell with 0.9 mL of MilliQ water. The resulting multivesicular liposomes were placed in an ice-cold water bath and sonicated three times each at 40% pulser and 40% power for 5 min using a 120 W sonicator probe (Ultrasonic Homogenizer Model 3000, Cary Biologics, Inc., Cary, NC, USA). A volume of 0.1 mL of 10X PBS was added to the 0.9 mL preparation to yield 1 mL liposomal formulation in 1X PBS. The preparation was analyzed and characterized prior to immunization.

To evaluate encapsulation efficiency (*EE* %), 250 µL of each formulation was transferred to 1 mL-thick walled polycarbonate ultracentrifuge tubes and diluted with 250 µL of PBS. The tubes were placed in an ultracentrifuge (Optima™ MAX-XP, Beckman Coulter, New South Wales, Australia) equipped with a TLA120.2 rotor. Centrifugation was conducted at 100,000 RPM for 1 hr at 4 °C. To determine the concentration, 100 µL of the supernatant was put through HPLC using the same method employed in the characterization of peptides (0 to 100% solvent B; C4 column). In parallel, 60 µL of each fresh formulation was diluted with 60 µL of PBS and put through the same HPLC protocol for a reference concentration. *EE* % values were calculated using Equation (1).
(1)$$EE\;\%=\left[1-\frac{Concentration\;\left(found\right)}{Concentration\;\left(reference\right)}\right]\times 100$$


**Size Distribution of Formulations**


A 200 µL volume of 10-fold PBS-diluted liposomal preparation was transferred to a disposable sample cuvette and the formulation size was evaluated using dynamic light scattering (Malvern, Worcestershire, UK) at 173 °C and 25 °C. Refractive indices were set to 1.45 and 1.332 for dispersed phase and dispersion medium (PBS), respectively. The viscosity of the dispersion medium was 0.9103 cp at 25 °C. Sample temperature was allowed to equilibrate for 5 min before measurement; collected data were set as the average of 11 runs, with 10 s duration per sample. Measurements that exceeded 100,000 counts per second were expected to yield representative and reproducibly sized data. Measurements were conducted in triplicate. Scatter data was analyzed using the general-purpose analysis model within the instrument’s software.


**Subcutaneous Immunization of Mice**


For initial screening studies, 6- to 8-week-old female BALB/c mice were randomly divided into eight groups of five mice. The mice were immunized subcutaneously, once with 50 µL/mouse containing 25 μg of CFA-adjuvanted B1, B2, B3, B4, B5, or a mixture of B1–B5 peptides. The peptide antigens were administered with equivalent masses, i.e. 25 μg of a given peptide antigen was loaded into 50 μL in volume of CFA formulation and injected into each mouse. The positive control group was immunized with CFA-adjuvanted RBD (Wuhan-1 original virus strain), S^319^–S^541^, at a dose of 25 µg in 50 µL/mouse, while the negative control group was injected with pure PBS. Blood was collected from each mouse before immunization (naïve), then again 14 days after immunization, via cardiac puncture. The blood was centrifuged at 10,000× *g* for 10 min to extract the serum. Sera samples were stored at −80 °C until further analysis.

To evaluate the vaccine candidates, 6- to 8-week-old female C57BL/6 mice were randomly divided into nine groups of five mice. Mice in the first six groups were immunized subcutaneously with 50 µL of vaccine formulation containing 25 μg of peptide antigen, as follows: LL-B1, LL-B2, LL-B3, LPMA-B3, B3-MF59, and L-B3-MF59. These peptide antigens were administered with equivalent masses, i.e. 25 μg of a given peptide antigen was loaded into a 50 μL volume of liposomal formulation and injected into each mouse. Another group was immunized with LL-B3 at a higher antigen dose of 50 µg in 50 µL/mouse. CFA-adjuvanted RBD (25 µg in 50 µL/mouse) was used to immunize the positive control group, and PBS was injected into mice in the negative control group. Mice were boosted 14 and 28 days after primary immunization, except for those in the CFA-adjuvanted RBD group, which were immunized only once. Naïve blood samples were collected before immunization and immune blood samples were collected on days 12, 26, and 42 after primary immunization via tail bleed. Sera were obtained and stored using the same approach as above.

Both studies were conducted according to regulations set by the National Health and Medical Research Council (NHMRC) of Australia (Australian Code of Practice for the Care and Use of Animals for Scientific Purposes). All animal procedures and protocols were approved by The University of Queensland Animal Ethics Committee (AEC), AEC Approval Number: SCMB/AIBN/069/17.


**Evaluation of Serological Immunogenicity**


Antipeptide sera samples were tested for immunogenicity and antibody avidity via enzyme-linked immunosorbent assay (ELISA) by coating either the same peptide or RBD, respectively. First, 96-well plates were coated with carbonate coating buffer containing 0.5 µg of peptide antigen or RBD. Plates were emptied, then blocked by coating the wells with 5% skim milk. The plates were emptied and washed with washing PBST buffer (PBS with 0.5% Tween 20). Serum samples were added to the plates and serially diluted in 0.5% skim milk starting with 100-fold dilutions, i.e., Log_10_ value of 2. The plates were emptied and washed three times with PBST buffer. Secondary antimouse IgG-Fc-HRP antibodies were diluted by adding 33 µL to 100 mL of 0.5% skim milk; 100 µL of this solution was then added to each well of the plates. The plates were emptied and washed three times with PBST buffer. O-phenylenediamine dihydrochloride substrate (OPD) was prepared and 100 µL was added to each well; this was left to react for 20 min at room temperature. The reaction was stopped through the addition of 50 µL of 1N H_2_SO_4_ solution to each well. The absorbance was measured at 450 nm wavelength using a SpectraMax microplate reader. Positive readings were tracked down the serial dilution of each serum sample until the response was not significant, i.e., the sum between the average and three times the standard deviation (3 × SD) of the naïve sera absorbance readings.


**Neutralization Efficacy Assays**


The neutralization efficacy of mouse serum was evaluated through various methods: ACE-2-RBD binding inhibition, pseudovirion neutralization, and live virus neutralization assays [49]. For facile initial screening of potentially neutralizing sera, an ACE2-RBD binding inhibition competitive ELISA-based method was employed, as reported previously [14,50].

Briefly, each well of a 96-well sample plate was coated with 100 µL of carbonate coating buffer containing 250 ng/mL of RBD. A separate calibration curve plate was also coated. Plates were blocked using 2% bovine serum albumin (BSA). Serum samples were added to group-designated sample plates, including the PBS group, and diluted in 0.5% BSA at a starting concentration of 5% dilution, followed by 2-fold serial dilutions four times down the plate. A volume of 100 µL of Fc-tagged human angiotensin converting enzyme (ACE2-Fc) solution (2.5 µg/mL) was added to each well of all sample plates. For the calibration curve plate, ACE2-Fc was serially diluted from 250 ng/well to 25 ng/well. Goat antihuman IgG-Fc-HRP secondary antibodies were added to all plates. OPD substrate was prepared and 100 µL was added to each well; the plates were left to react for 20 min at room temperature. The reaction was stopped by adding 50 µL of 1N H_2_SO_4_ solution to each well. Absorbance was measured at 450 nm wavelength using a microplate reader (SpectraMax 250, Molecular Devices, San Jose, CA, USA). To determine the content of bound or missing ACE2 in each well, calibration curves (*n* = 4) were fit to logistic function (R^2^ ≈ 0.98), Equation (2). The best fit equation was used with absorbance (*y*) values of each well to determine bound ACE2 content (*x*) by interpolation. The inhibition percentage was determined from the missing amount of ACE2, Equation (3). Where *y* is the absorbance reading, *A_1_* and *A_2_* are minimum and maximum absorbance readings, respectively; *x* is ACE2 concentration in ng/100 µL/well; *x*_0_ is the x-axis center point; *p* is the power exponent; and the ACE2_total_ value is 250 ng/100 µL/well.
(2)$$y=\frac{A_{1}-A_{2}}{1+{\left(x/x_{o}\right)}^{p}}+A_{2}\;$$
(3)$$Inhibition\;\%=\frac{ACE2_{Total}-ACE2_{Bound}}{ACE2_{Total}}\times 100$$

Pseudovirion neutralization assay was conducted as described previously [50,51]. Briefly, neutralization was measured against a pseudotyped virus that encoded the spike glycoprotein, which was produced via lipofectamine transfection of HEK-293T cells. The SARS-CoV-2 pseudotyped virus was based on the HIV-1 lentiviral packaging system incorporating luciferase reporter. The HEK293T were transduced to produce human ACE-2 by lentivirus. Luciferase activity of virus stock was determined in ACE2-HEK293T by measuring relative luciferase unit infectious dose. Heat-inactivated immune sera were serially diluted 5-fold in Dulbecco’s Modified Eagle Medium (DMEM) and mixed with 50,000 RLU of virus, then incubated for 1 h at 37 °C. The serum–virus mixtures were added to the seeded ACE2-HEK293T cells (1 × 10^4^) in a 96-well plate in a total volume of 150 µL, and the cells were incubated for 72 h at 37°C and 5% CO_2_. After incubation, medium volume was aspirated and replaced with 100 µL of BioGlo^TM^ (luciferase enzyme substrate, Promega, Cambridge, UK) and left for 5 min to lyse the cells and allow luciferase enzyme reaction. Finally, a volume of 100 µL of cell lysate was transferred to a black-walled plate for luminescence measurement using a microplate reader (Varioskan^TM^ Thermo Fisher Scientific, Waltham, MA, USA). Samples were tested in duplicate, and neutralization values were determined as the percentage RLU of positive (immune serum-free) virus control. Neutralization efficacy is presented as the reciprocal serum dilution that achieved 50% neutralization compared to controls (luciferase fluorescence intensity).

Live virus neutralization assays (PRNTs) were conducted as described previously [49,52]. Briefly, block buffer, wash buffer (PBST), and overlay medium were prepared, and mouse serum was heat-inactivated. SARS-2 virus isolate (QLD02/2020 as D614, GISAID accession EPI_ISL_407896, and QLDID935/2020 as G614, GISAID accession EPI_ISL_436097 provided by Queensland Health Forensic and Scientific Services, Queensland Department of Health, Queensland, Australia) was used for the assay. Vero E6 (5 × 10^4^) cells were seeded in 96-well plates with DMEM medium and incubated overnight at 37 °C and 5% CO_2_. After incubation, the medium was discarded, mouse serum was serially diluted 5-fold across the plate, and 260 FFU of viral inoculum was added to each well and left to incubate for 1 h at 37 °C. In a similar process, as a control, a concentration of 10 µg/mL of S309 neutralizing antibodies was also serially diluted 5-fold, then incubated with the same inoculum of SARS-CoV-2 virus and the seeded Vero E6 cells. The overlay medium was added to the cells, and the plates were re-incubated for 14 h at 37 °C and 5% CO_2_. The cells were then fixed with 80% acetone. The plates were dried and blocked using blocking buffer [milk diluent sera (KPL, Seracare) and 0.1% Tween in PBS] for 1 h at room temperature. The plates were probed with antispike antibody (CR3022), then IR dye^®^-conjugated secondary antibodies (LI-COR Biosciences, Lincoln, NE, USA), both diluted in blocking buffer by addition to each of the seeded cell wells. The plates were read using an Odyssey Infrared Imaging System infrared high-resolution scanner LI-COR CLX (LI-COR Biosciences, Lincoln, NE, USA). Spots denoting the number of infected Vero E6 cells were counted using ImageJ Fiji software version 1.53e (a free, open-source application).


**Statistical Analysis**


Parametric one-way ANOVA with Dunnett’s multiple comparison test was used to evaluate all assays. Significance was set at *p* < 0.05 (*), *p* < 0.01 (**), *p* < 0.001 (***), and *p* < 0.0001 (****).

## 3. Results and Discussion

The RBD sequence (S^329^–S^421^) encloses several critical binding residues essential for binding to human ACE2 receptors [35]. In particular, the RBM peptide sequence S^444^–S^508^ is in intimate contact with the ACE2 receptor [31]. Altering a single critical binding residue greatly diminishes ACE2 binding, thus effectively neutralizing the spike protein fusion process. Therefore, in this study, we investigated two long sequences that bracket the RBM sequence: B2 (S^469^–S^508^) and B3 (S^444^–S^483^). In addition, peptide B5 (S^366^–S^395^), which encloses the SARS-CoV-conserved epitope footprint of moderately neutralizing antibody CR3022 [53], was also considered. The purpose of employing such long sequences was to preserve the native structural conformation of these derived peptides, since the enclosed neutralizing epitope is potentially discontinuous. Moreover, we included two peptides that could potentially, upon neutralization, disrupt the priming process: B1 (S^623^–S^639^) and B4 (S^559^–S^572^); these are located at or near the peptide priming site that separates the S1 and S2 subunits of the spike protein.

Peptides derived from the RBM region were designed to be longer than typical B-cell epitopes to maintain their structural conformation [35], to screen them as vaccine antigens, and to evaluate their in vitro neutralization efficacy. Since CFA is considered the gold-standard adjuvant due to its powerful immunostimulatory effect, it was employed for peptide screening to avoid the impact of a weak adjuvant on the immunogenicity and efficacy of the antigens. Peptides B1–B5 were formulated with CFA adjuvant (individually and combined) and compared to CFA-adjuvanted RBD for the ability to stimulate antibody production following subcutaneous BALB/c mice immunization (25 µg peptide antigen in 50 µL/mouse formulation). Serum was collected from the mice and evaluated for its immunogenicity, avidity, and neutralization efficacy using ELISA and cell-based assays.

The most promising peptide antigens, B1–B3, were then conjugated to polyleucine (L_10_) or PMA, anchored to the liposomes, characterized, and used to subcutaneously immunize C57BL/6 mice. The conjugates were compared to CFA-adjuvanted RBD as a positive control. The clinically approved adjuvant MF59^®^ was also employed to formulate promising peptide antigens. Mouse serum was collected to assess immunogenicity and avidity via antipeptide and anti-RBD ELISA assays, and to assess neutralization efficacy using competitive ELISA as a quick screening tool. This was followed by more in-depth evaluation of the promising groups’ neutralization efficacy using pseudovirion neutralization assays and live virus (plaque reduction) neutralization assays.

### 3.1. Secondary Structure of Peptide Antigens

Circular dichroism was employed to evaluate the secondary structure of the peptides and their conjugates. Peptides B1–B5 adopted mostly random coil structures (Appendix A and Figure 2A). Random coil conformation content for B1–B4 ranged from 60–70%. To a lesser extent, B1, B3, and B4 partially adopted α-helical conformations (27–29%), whereas B2 showed 18% α-helical conformation. B2 and B3 secondary structures aligned with the reported cryo-EM spike protein secondary structure measurements (Appendix A) [1]. In contrast, B5 predominantly adopted β-sheet conformation (60%, compared to the original conformation of ca. 13%) at the expense of random coil content. In addition, B4 included α-helical conformation (30%), whereas the native peptide adopted only β-sheet conformation (18%) and no α-helical conformation. Therefore, B3 and B2 were expected to generate antibodies with higher avidity, i.e., recognition of parent protein with native conformation, compared to the other peptides.

Notably, conjugation of L_10_ to B2 and B3 in the follow up study for adjuvant screening changed the structural conformation of these peptides dramatically to result in much higher α-helical conformation contents of 60% and 95%, respectively (Figure 2B). Thus, the incorporation of L_10_ into L_10_-B2 and L_10_-B3 altered the native dominant random coil conformation, as the helical content far exceeded that which would account for the additional leucine residues in the sequence. However, the conjugation of L_10_ to B1 resulted in the adoption of special all-β-protein-like structures [54]. This has been reported previously for several antigens adjuvanted with polyleucine moieties [55]. This suggests that these peptide antigens would not produce antibodies with high avidity, unless inclusion into liposomes restored the secondary structure to native conformation, as has been reported to occur when peptides were present in the proximity of lipid bilayers or emulsion oil globules [56,57,58].

### 3.2. Size and Encapsulation of the Formulations

In the initial peptide antigen screening study, compounds B1–B5 were synthesized without hydrophobic adjuvating moieties; thus, they were free, water-soluble peptides formulated in CFA water-in-oil emulsion adjuvant. However, the peptide adjuvant screening study involved the conjugation of peptide antigens to hydrophobic moieties, such as PMA and L_10_, and anchoring to emulsion or liposomes. This typically facilitates the antigen presentation process and, consequently, increases immunogenicity [36,40,59,60,61,62,63]. The encapsulation efficacies were high (≥80%) for all liposomal preparations.

All liposomal formulations formed mostly small particles (120–150 nm); however, larger aggregates (microparticles) were also detected (Appendix A). The polydispersities of the liposomes were low (PDI ≤ 0.2), apart from LL-B3 (50 µg dose, PDI = 0.355 ± 0.017) and LPMA-B3 (PDI = 0.528 ± 0.0134). The presence of large aggregates in liposomal formulations (~1.5 µm) suggested that a small fraction of conjugated peptide antigens was not encapsulated (Appendix A). This observation was further confirmed by measurement of the L-B1, L-B2, and L-B3 encapsulation efficacy, which showed that ≤20% of conjugated peptides were free/nonencapsulated in the liposomal formulations.

### 3.3. Immunogenicity of Peptide Antigens and Self-Adjuvanted Peptide Vaccines

CFA is considered the most powerful, gold-standard adjuvant; thus, it was chosen for peptide antigen screening in mice (Figure 3). Mice treated with CFA-adjuvanted B3 and RBD produced significantly higher anti-RBD IgG levels in the serum compared to the negative control group (*p* < 0.001 and *p* < 0.0001, respectively) following single subcutaneous immunization (Figure 3B). None of the other peptides triggered significant serum anti-RBD IgG titers (*p* > 0.05). B3–B5-immunized mice produced antibodies that recognized B3-B5 peptide epitopes (Figure 3C), but neither B4 nor B5 recognized RBD coated on ELISA plates (Figure 3B). B1- and B2-immunized mice did not produce antibody responses, even against B1 and B2 peptides. Antibody titers produced by B3 against itself were substantially higher than those produced by any other peptides, and most importantly, the antibodies recognized RBD. Thus, the B3 peptide antigen most likely adopted the correct structural conformation and was chosen for further investigation.

B1 and B2 were included in the follow up adjuvant screening studies to act as (potential) negative controls. This was considered pertinent, as the mutation N^501^Y located in the B2 (S^469^–S^508^) sequence that is present in alpha, beta, and gamma lineages reduced the efficacy of the ChAdOx-1 nCoV-19 (AZD1222, Astra Zeneca) DNA vaccine from 76% against the original strain to 22% against the N^501^Y strains, with particular variation against the beta variant (B.1.351) [64]. However, CFA-RBD-immunized mice generated predominantly anti-B2 and anti-B3 IgG, showing that there are potentially neutralizing antibodies with a partial or complete epitope footprint on the B2 sequence. B1 was excluded only because of its low immunogenicity; however, repeated administration with an L_10_/liposome system may boost its ability to stimulate antibody production. Moreover, convalescent patient serum was found to contain antibodies against the B1 peptide, which implies that it may also contain a partial or complete neutralizing epitope within its short sequence, e.g., S^475^–S^499^ [65]. Consequently, B2 and B3 were further progressed to peptide vaccine adjuvant screening studies.

All mice were immunized three times, except for those in the CFA-adjuvanted RBD group, which were immunized only once. The liposome formulations with L_10_- or PMA-conjugated peptide antigens proved to be powerful adjuvants for peptide vaccines [34,39,40]; thus, they were employed to help maximize the immunogenicity of B1–B3 peptides. All formulations contained clinically (human) tested PADRE as a universal T-helper epitope [66]. C57BL/6 mice were immunized with liposome formulations: LL-B1 (25 µg), LL-B2 (25 µg), or LL-B3 (25 µg or 50 µg), as well as LPMA-B3 (25 µg). Two additional groups were immunized with either B3-MF59 (25 µg), or CFA-adjuvanted RBD (25 µg): the latter was used as a positive control group.

B3-MF59 and LL-B3 (50 µg) induced the highest anti-B3 IgG titers (log_10_ = 3.75 and 3.3, respectively) following single dose immunization (Appendix A). Further, anti-B3 titers of B3-MF59 were similar to those of the CFA-immunized group (*p* > 0.05). After the second immunization, anti-B3 IgG titers were high: log_10_ IgG titers of 5–6 for CFA-adjuvanted RBD, B3-MF59, LL-B3 (50 µg), and LL-B3 (25 µg), similar to those induced by CFA-adjuvanted RBD mice (*p* <0.05). Notably, neither the B2 nor B1 vaccines generated antipeptide IgG titers in the serum after two immunizations. CFA-adjuvanted RBD mouse serum generated significant anti-B2 IgG titers (log_10_ = 3, *p* < 0.0001) in the serum (Appendix A).

All B3-immunized mice generated high anti-B3 IgG titers following the third immunization, comparable to CFA-adjuvanted RBD mice (*p* > 0.05), except for LB3-MF59 (Figure 4A). Mice immunized with LL-B2 also generated anti-B2 antibody titers (log_10_ = 3) similar to anti-B2 IgG titers of the CFA-adjuvanted RBD group (*p* > 0.05) (Figure 4C). Furthermore, B3- and B2-immunized mice generated IgG antibodies in the serum that recognized RBD (Figure 4B). However, the level of LL-B2-generated anti-RBD-IgG antibodies was close to the detection limit of ELISA. B3-MF59, LL-B3 (25 or 50 µg), and LPMA-B3 produced the highest levels of anti-RBD IgG titers; their levels were similar to those induced by CFA-adjuvanted RBD. This demonstrated high avidity of generated anti-B3 antibodies when the peptide is anchored to liposomes. However, LL-B2- and L-B3-MF59-immunized mice generated significantly lower serum anti-RBD IgG titers compared to RBD-immunized mice (*p* < 0.0001).

Generally, liposome/L_10_ and MF59 (with soluble antigens) induced IgG titers as high as CFA-adjuvanted RBD after the first immunization. The antibodies produced had high avidity, recognizing RBD coated on ELISA plates. However, the addition of L_10_-conjugated peptide antigen to MF59 reduced immunogenicity significantly, likely due to the peptides self-assembling separately from the emulsion oil globules (Appendix A).

### 3.4. Neutralization Efficacy

Immunized mouse sera were evaluated for its neutralization efficacy using competitive ELISA [50] and pseudovirion neutralization assays [51,52,67,68]. CFA-adjuvanted B3-immunized sera (from the first study, Figure 3) demonstrated high and persistent neutralization/inhibition (ca. 70%) of ACE2/RBD binding in competitive inhibition assays (Figure 5), down to 1/80 serum dilution, which was comparable to sera from mice immunized with CFA-adjuvanted RBD (*p* > 0.05). CFA-adjuvanted B3 sera was also able to neutralize ACE2/RBD binding in the second immunization experiment; however, the neutralization capacity was lower than in the first experiment. This may reflect the fact that different breeds of mice were used between the experiments. B3-MF59-immunized sera were the most neutralizing of those tested in the second study and were comparable to the CFA-adjuvanted RBD group; however, variability between mice within both groups was very high. The neutralization/inhibition efficacy of B3-MF59, LPMA-B3, and LL-B3 sera dropped swiftly with dilution; 50% inhibition was achieved at serum dilutions of ≤1/40.

The pseudovirion neutralization test assesses the neutralization efficacy of immune sera without the need to use pathogenic live virus, eliminating the rigorous containment requirements to conduct the assay, while still maintaining high relevance to real infection processes in vivo (Figure 6). Thus, immune sera from selected mouse groups were evaluated using pseudovirion neutralization assays. Resulting neutralization capacities were generally in agreement with the serum’s ability to inhibit RBD/ACE2 binding. CFA-adjuvanted B3 sera were highly neutralizing (nAb titers of 296) and comparable to that of CFA-adjuvanted RBD (nAb titers of 195, *p* > 0.05). However, sera from C57BL/6 mice in the adjuvant screening study did not significantly neutralize the virus compared to PBS group (*p* > 0.05) and were far less neutralizing compared to the first study (involving BALB/c mice).

Alum-adjuvanted RBD was recently reported to be weakly neutralizing in C57BL/6 mice [68], even though CFA-adjuvanted RBD vaccine was reported to be highly neutralizing in several studies (and in our initial antigen screening study with BALB/c mice) [16]. The differences in RBD/ACE2 binding inhibition and pseudovirion neutralization ability between CFA-adjuvanted RBD groups from the two experiments suggests that the change in mouse strain resulted in the heterogeneity of responses. Moreover, the neutralization capacity of mouse serum from the adjuvant screening study was further confirmed with live virus (plaque reduction) neutralization assays (Appendix A). In all three assays (competitive ELISA, pseudovirion neutralization, and live virus neutralization), serum from mice in the second study had moderate to low neutralization efficacy with nAb titers < 60. Neutralization was most effective in the CFA-adjuvanted RBD group, followed by B3-MF59 and LPMA-B3.

L_10_/liposomes, PMA/liposomes, and MF59′s suitability as adjuvants and delivery systems for peptide antigens may depend on several factors, including how an enclosed neutralizing epitope is presented to or recognized by B-cell receptors. Modification of the *N*-terminal fragment of peptide antigen (B3) with polyleucine or PMA, as well as anchoring of conjugates to liposomes, may reduce the accessibility of the *N*-terminus fragment of the antigen to the environment and recognition by B-cell receptors. Recently, mice immunized with CFA-adjuvanted peptides derived from spike protein were explored for potential neutralizing epitopes [38,65]. Several RBD-derived epitopes were found to be moderately neutralizing, e.g., S^439^–S^454^ and S^455^–S^469^, with nAb titers ranging from 30 to 60 in pseudovirion neutralization assays [65]. Low levels of neutralizing antibodies were produced, most likely because the tested epitopes were short 15-mer peptides. The neutralizing epitopes tested (S^439^–S^454^, S^455^–S^469^, or S^435^–S^479^) overlap/coincide with the *N*-terminus of B3 (S^444^–S^483^) [38,65]. This may explain the low neutralization capacity of *N*-terminus-anchored B3 epitopes. Nonetheless, CFA-adjuvanted B3 triggered the production of highly neutralizing antibodies, equal to or greater than those from immunization with RBD/CFA (*p* > 0.05). Thus, alternative orientation/anchoring strategies of B3 epitopes, e.g., via conjugating the L_10_ moiety to the *C*-terminus of B3, may improve the efficacy of the conjugates. However, synthesis of such compounds could be very challenging.

In summary, we identified a promising peptide-based SARS-CoV-2 vaccine antigen (B3). However, to produce effective B3-based vaccine alternatives, powerful adjuvant systems are required as CFA is not safe for humans, and the alternative adjuvants tested in this study were not adequately effective. Furthermore, the introduction of a helix or strand promoting adjuvant moieties altered the secondary structure of the peptides, and potentially rendered them inadequately neutralizing. It would be interesting to evaluate whether similar effects would still occur if peptide antigens were conjugated to random coil-supporting poly(hydrophobic amino acids), for example: polyproline or polyaspartic acids [69]. Alternatively, as CFA emulsion was effective in the current study, future research can focus on clinically approved water-in-oil emulsion adjuvants, such as IFA and montanide ISA (for clinical trials) as a substitute for the toxic, albeit powerful, CFA adjuvant.

## 4. Conclusions

We identified a new SARS-2-S B-cell epitope. B3 (S444–S483) proved to be the most immunogenic peptide antigen tested. CFA-adjuvanted B3-immunized mice produced sera with significant neutralization efficacy against SARS-CoV-2. Adjuvating systems relying on liposomal formulations of polyleucine or polymethyl acrylate increased the immunogenicity of B3; however, they did not help to achieve similar neutralization efficacy. Emulsion-based adjuvants, such as CFA and MF59, improved the avidity of the antibodies generated and their neutralization efficacy against SARS-CoV-2. Further effort to optimize peptide antigen delivery is required.

## Figures and Tables

**Figure 1 pharmaceutics-14-00856-f001:**
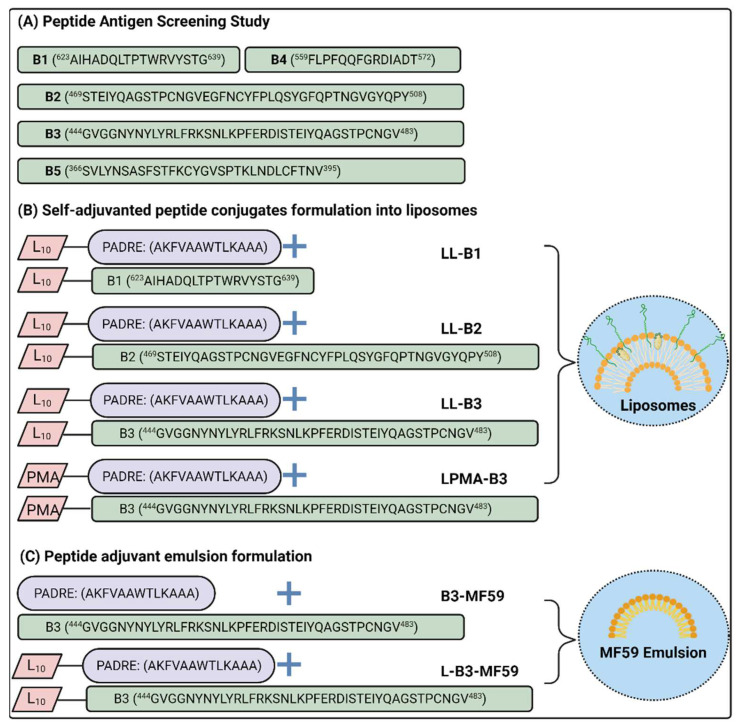
SARS-2 peptide antigens and their vaccine formulations. Amino acid single code letters were used to describe the synthesized sequences. (**A**) Peptide antigens B1–B5 and their location in the spike protein sequence; (**B**) self-adjuvanted peptide–polymer conjugates formulated into liposomes; and (**C**) peptide antigen, B3, and its polyleucine conjugate emulsified with MF59. L refers to 10-mer polyleucine; LL refers to liposome coated with polyleucine-conjugated peptide for anchoring; LPMA refers to liposome coated with PMA-conjugated peptide antigen; MF59 is an emulsion adjuvant formulation; PADRE is universal helper T-cell epitope; and PMA is polymethyl acrylate polymer.

**Figure 2 pharmaceutics-14-00856-f002:**
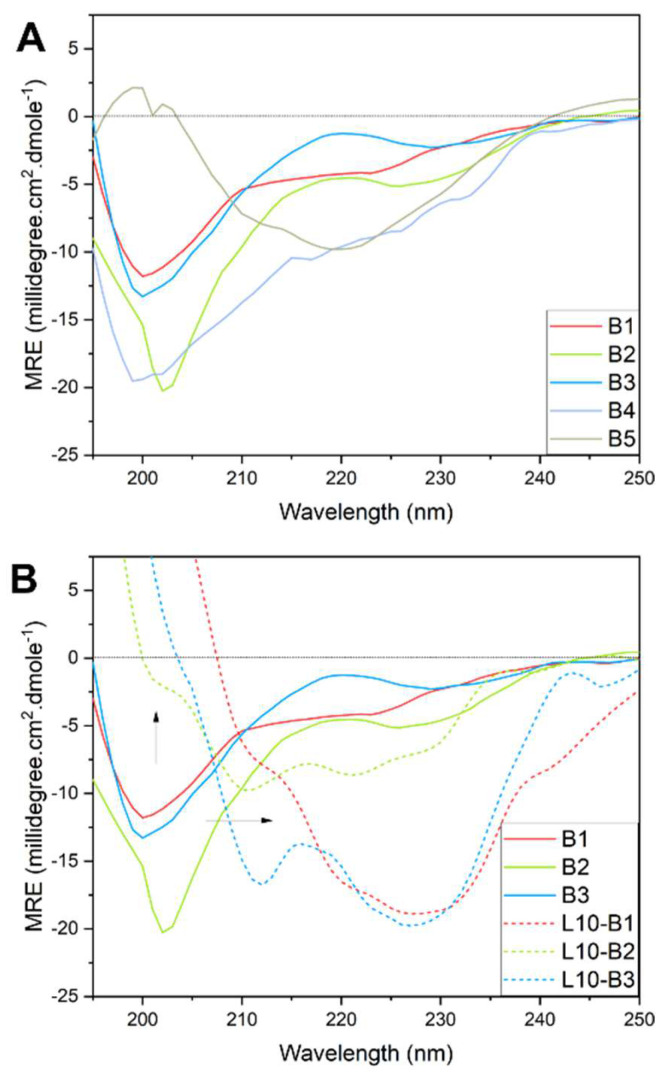
Circular dichroism measurements of the secondary structures of (**A**) the peptide antigens and (**B**) their polyleucine conjugates. The arrows indicate the shift from 197 nm wavelength minima, which denote random coil structure conformation, to higher wavelength minima that show β-sheet and α-helical structures. (**A**) Nonconjugated peptides B1-B4 adopted predominantly a random coil structure conformation 60–70%, with small α-helical contents <30%. B5 peptide adopted predominantly β-sheet conformation (60%), despite that the original peptide sequence in the protein has only 13% β-sheet conformation content (Appendix A). (**B**) Conjugation of B1–B3 to polyleucine have dramatically altered secondary structures, compared to native conformation, with increased α-helical content ranging from 60–95%.

**Figure 3 pharmaceutics-14-00856-f003:**
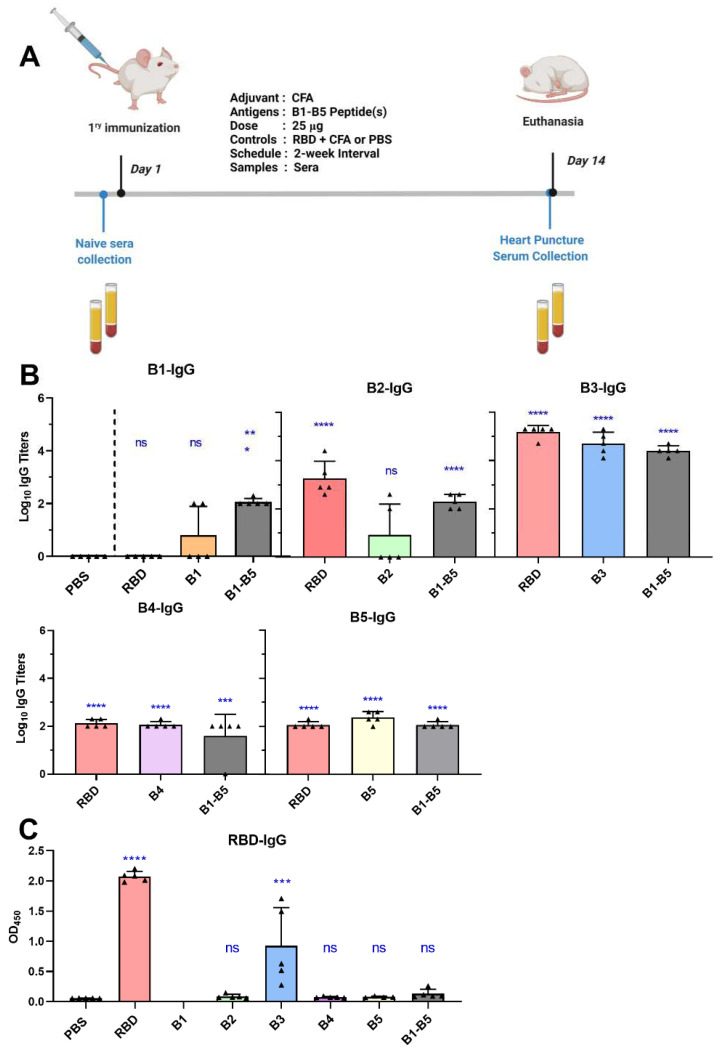
Antigen screening study in BALB/c mice. (**A**) Immunization conditions and schedule. (**B**) Antigen-specific serum IgG titers in mice immunized with CFA-adjuvanted peptide antigens. Antibody titers have been measured by ELISA with the plates coated with B1-B5 antigens to measure levels of B1-IgG, B2-IgG, B3-IgG, B4-IgG, and B5-IgG, respectively. (**C**) Anti-RBD IgG concentration/absorbance in the sera of mice immunized with CFA-adjuvanted peptide antigens. Triangles describe individual mouse values in each group. Error bars represent the standard deviation. Statistical analysis was performed using one-way ANOVA compared to the negative control (PBS) group with Dunnett’s multiple comparison test. (ns) nonsignificant, *p* > 0.05; (*) *p* < 0.05; (**) *p* < 0.01; (***) *p* < 0.001, and (****) *p* < 0.0001.

**Figure 4 pharmaceutics-14-00856-f004:**
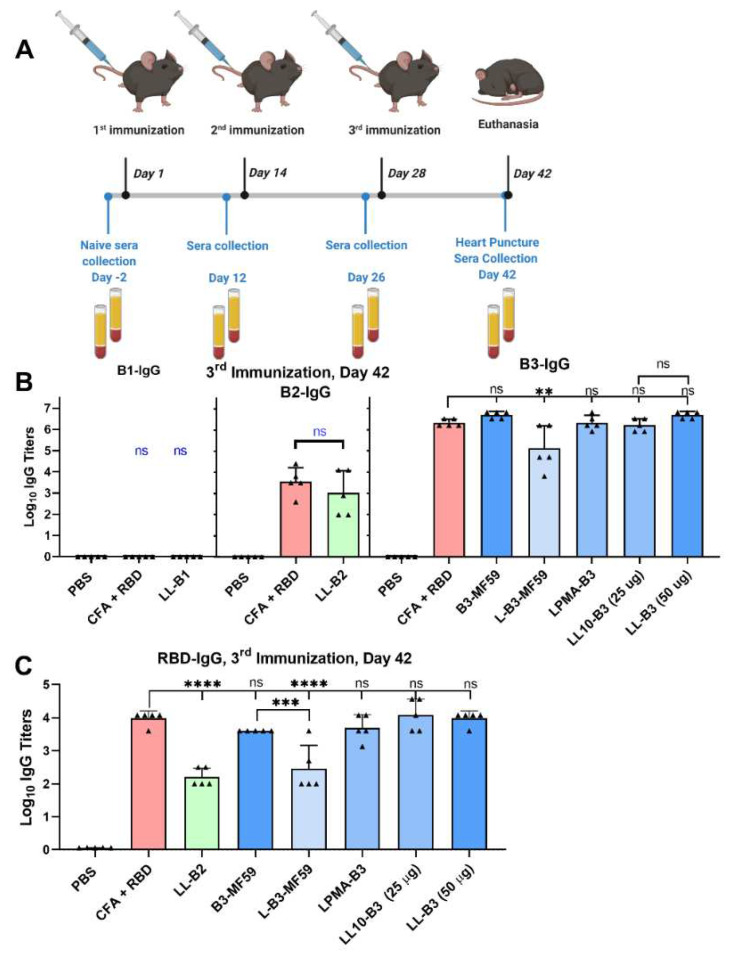
Adjuvant screening study in C57BL/6 mice. (**A**) Mice immunization schedule. (**B**) Antigen-specific serum IgG titers in mice immunized with liposomal formulation of conjugated antigens (LL-B1, LL-B2, LL-B3, LPMA-B3) and emulsion-based formulations (B3-MF59 and L-B3-MF59). Antibody titers have been measured by ELISA with the plates coated with B1-B3 antigens to measure levels of B1-IgG, B2-IgG, and B3-IgG, respectively. (**C**) Anti-RBD IgG titers in the serum after three immunizations. Triangles describe individual mouse values in each group. Statistical analysis was performed using one-way ANOVA with Dunnett’s multiple comparison test. (ns) nonsignificant, *p* > 0.05; (**) *p* < 0.01; (***) *p* < 0.001, and (****) *p* < 0.0001.

**Figure 5 pharmaceutics-14-00856-f005:**
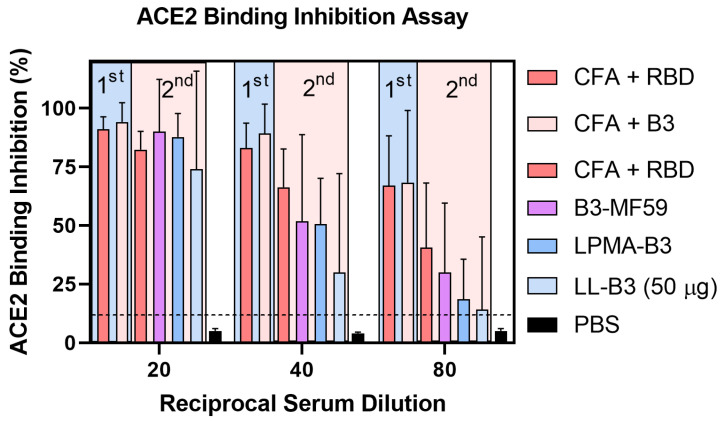
Competitive ELISA binding inhibition assays of immune sera, where ability of antibodies from sera of immunized mice to inhibit interaction between ACE2 and RBD of spike protein was measured. The inhibition efficacies of antibodies collected from sera of mice immunized in the antigen screening study are highlighted in blue, while ones from the adjuvant screening study are highlighted in light red. The mice sera have been diluted 20, 40, and 80 times.

**Figure 6 pharmaceutics-14-00856-f006:**
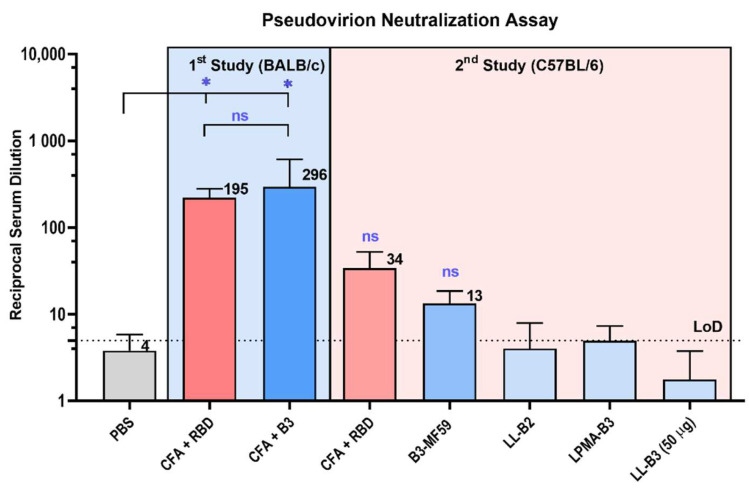
Pseudovirion neutralization assay of mice sera. The neutralization efficacy of antibodies collected from sera of mice immunized in the antigen screening study are highlighted in blue, while ones from the adjuvant screening study are highlighted in light red. Neutralization was measured against a pseudotyped virus that encoded the spike glycoprotein, which was produced via lipofectamine transfection of HEK-293T cells. Samples were tested in duplicate, and neutralization values were determined as the percentage RLU of positive (immune serum-free) virus control. Neutralization efficacy is presented as the reciprocal serum dilution that achieved 50% neutralization compared to controls (luciferase fluorescence intensity). CFA-adjuvanted B3 peptide generated the highest mean neutralizing antibody titers level of 296, CFA-adjuvanted RBD generated mean neutralizing antibody titers of 195 (these numbers are placed close to the relevant bars on the figure). Statistical analysis was performed using one-way ANOVA with Dunnett’s multiple comparison test. (ns) nonsignificant, *p* > 0.05; (*) *p* < 0.05.

## Data Availability

Not applicable.

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
