# Peer review of "Peptide-Based Vaccine against SARS-CoV-2: Peptide Antigen Discovery and Screening of Adjuvant Systems"

_pharmaceutics, 2022, doi:10.3390/pharmaceutics14040856_

Round 1

Reviewer 1 Report

The manuscript titled "Peptide-based Vaccine against SARS-CoV-2: Peptide Antigen Discovery and Screening of Adjuvant Systems" is focussed on  peptides that are derived from SARS-COV-2 for neutralizing B-cell peptide antigens. The results are quite interesting. The authors identified B3 as potent vaccine antigen among the tested. The methods are clearly described for synthesis peptide antigens and further analysis. 

The result and conclusions are well supported by the experimental data. Overall, the manuscript is well written with excellent design of experiments. The content of the manuscript is timely needed. I am certain that the manuscript will be of interest to wide range of readers.

Minor suggestion: the length of abstract could be condensed. 

Author Response

The manuscript titled "Peptide-based Vaccine against SARS-CoV-2: Peptide Antigen Discovery and Screening of Adjuvant Systems" is focussed on peptides that are derived from SARS-COV-2 for neutralizing B-cell peptide antigens. The results are quite interesting. The authors identified B3 as potent vaccine antigen among the tested. The methods are clearly described for synthesis peptide antigens and further analysis. 

The result and conclusions are well supported by the experimental data. Overall, the manuscript is well written with excellent design of experiments. The content of the manuscript is timely needed. I am certain that the manuscript will be of interest to wide range of readers.

Minor suggestion: the length of abstract could be condensed. 

Response: We have condensed the abstract from 350 words to 231 words, as follows:

“The SARS-CoV-2 virus has caused a global crisis, resulting in 0.5 billion infections and over 6 million deaths, as of March 2022. Fortunately, infection and hospitalization rates were curbed due to the rollout of DNA and mRNA vaccines. However, the efficacy of these vaccines significantly drops a few months post immunization: from 88% down to 47% in the case of the Pfizer BNT162 vaccine. The emergence of variant strains, especially delta and omicron, have also significantly reduced vaccine efficacy. We propose peptide vaccines as a potential solution to address the inadequacies of the current vaccines. Peptide vaccines can be easily modified to target emerging strain, has greater stability and do not require cold-chain storage.  We screened five peptide fragments (B1–B5) derived from the SARS-CoV-2 spike protein to identify neutralizing B-cell peptide antigens. We then investigated adjuvant systems for efficient stimulation of immune responses against the most promising peptide antigens, including liposomal formulations of polyleucine (L10) and polymethylacrylate (PMA), as well as classical adjuvants (CFA and MF59). Immune efficacy of formulations was evaluated using competitive ELISA, pseudovirion neutralization, and live virus neutralization assays.  Unfortunately, peptide conjugation to L10 and PMA dramatically altered the secondary structure, resulting in low antibody neutralization efficacy. Of the peptides tested, only B3 administered with CFA or MF59 was highly immunogenic. Thus, a peptide vaccine relying on B3 may provide an attractive alternative to currently marketed vaccines.”

Reviewer 2 Report

Generally interesting, well-explained, and helpful manuscript. Minor comments: 

1) The introduction to self-adjuvanting moieties could be expanded a bit more as it is one of the critical designs of the vaccine. At the moment, the reader is forced to dive into the references to obtain more information (shouldn't have to). 

2) It would be helpful to visualise the 3D locations of the B1-5 peptides on the spike protein. The rationale of the selection could also be better explained. 

3) In addition to the weight (mg), it's also best to include the mmol information for the liposomal formulations for both the lipids and peptides so that the reader won't have to hand calculate the actual formulation compositions. 

4) The circular dichroism (CD) measurement and the interpretation of secondary structure needs to be better explained in text and annotated on figures. In addition, captions need to be expanded. 

5) Most figures lack descriptive captions to guild readers through the data. 

6. Size distribution, does it matter for the current vaccine as the distribution is very polydispersed?

Author Response

Please attached response to reviewer comments (including Reviewer #2)

Reviewer 3 Report

This paper has identified a potent epitope B3, for which CFA adjuvanted promoted significant neutralization efficacy against SARS-CoV-2. This worked screened this antigenic peptide, by detail investigating the antibody immune responses in the mouse models. The manuscript was well presented and the collected data clearly indicated the efficacy of B3-CFA is comparable to  that of RBD + CFA. I suggest to accept this manuscript and provide some comments for the author to consider in their revision of the manuscript.  

  1. Does the dose (25 ug per mouse) mean the mass of antigen only or LL-B1/B2/B3 etc? As the molecular weight is different for different antigens, whether the molar concentration is better for comparison?
  2. Is the control RBD the full length of COVID-19 virus’s RBD? Which strain is it belonged to?
  3. Fig 3: did you measure the particle size distribution of liposome without loading peptide? Monodispersed? Why is there a big particle band? Aggregation?
  4. B3 was found to be the most potent. Whether B3 is in the key sequence region of RBD and has stable secondary structure?

Author Response

Please attached response to reviewer comments (including Reviewer #3)

Reviewer 4 Report

The research focused on Peptide-based Vaccine against SARS-CoV-2: Peptide Antigen Discovery and Screening of Adjuvant Systems with Liposomes. Generally speaking, the work is well planned and the data are conceiving with the hypothesis.

Minor Corrections:

  1. The resolution quality of all figures needs to be improved; especially figure 1.
  2. However, I am curious; why authors have used liposomal system for their work, as there are other delivery systems like Dendrimers; which can provide better conjugation facility for modification of the delivery system which can explains exact numbers of molecules conjugated per dendrimer surface.
  3. I suggest authors to compare the particle size of their Adjuvant Systems by any one other methods like FE-SEM, HR-TEM or AFM. The data presented by DLS won’t be sufficient to proof the concept.

Author Response

Please attached response to reviewer comments (including Reviewer #4)
